# On the Fused Deposition Modelling of Personalised Bio-Scaffolds: Materials, Design, and Manufacturing Aspects

**DOI:** 10.3390/bioengineering11080769

**Published:** 2024-07-31

**Authors:** Helena Cardoso Sousa, Rui B. Ruben, Júlio C. Viana

**Affiliations:** 1IPC/LASI—Institute of Polymers and Composites/Associated Laboratory in Intelligent Systems, Polymer Engineering Department, University of Minho, 4800-058 Guimarães, Portugal; jcv@dep.uminho.pt; 2ESTG-CDRSP, Polytechnic Institute of Leiria, 2411-901 Leiria, Portugal; rui.ruben@ipleiria.pt

**Keywords:** additive manufacturing, 3D printing, fused deposition modelling, bone tissue engineering, bio-scaffold, biocompatible, biodegradable

## Abstract

Bone tissue engineering (BTE) is an important field of research, essential in order to heal bone defects or replace impaired tissues and organs. As one of the most used additive manufacturing processes, 3D printing can produce biostructures in the field of tissue engineering for bones, orthopaedic tissues, and organs. Scaffold manufacturing techniques and suitable materials with final structural, mechanical properties, and the biological response of the implanted biomaterials are an essential part of BTE. In fact, the scaffold is an essential component for tissue engineering where cells can attach, proliferate, and differentiate to develop functional tissue. Fused deposition modelling (FDM) is commonly employed in the 3D printing of tissue-engineering scaffolds. Scaffolds must have a good architecture, considering the porosity, permeability, degradation, and healing capabilities. In fact, the architecture of a scaffold is crucial, influencing not only the physical and mechanical properties but also the cellular behaviours of mesenchymal stem cells. Cells placed on/or within the scaffolds is a standard approach in tissue engineering. For bio-scaffolds, materials that are biocompatible and biodegradable, and can support cell growth are the ones chosen. These include polymers like polylactic acid (PLA), polycaprolactone (PCL), and certain bioglass or composite materials. This work comprehensively integrates aspects related to the optimisation of biocompatible and biodegradable composites with the low cost, simple, and stable FDM technology to successfully prepare the best designed composite porous bone-healing scaffolds. FDM can be used to produce low-cost bone scaffolds, with a suitable porosity and permeability.

## 1. Introduction

Bone tissue is a complex anisotropic material with pivotal roles in the human body, including locomotion, organ protection, and mineral storage [1]. Bone fractures are the most common traumatic injuries in humans [2], while chronic bone defects due to osteoporosis, tumor removal, traumatic incidents, and congenital abnormalities present ongoing clinical hurdles [3]. As global populations rise, there is also a noticeable increase in bone degenerative diseases. In 2017, approximately 22.3 million orthopaedic surgeries were performed worldwide, a number expected to grow to 28.3 million by 2022 [4]. Each year, over 2.2 million individuals globally require bone graft surgeries to repair significant bone defects [5].

The loss of bone significantly detracts from patients’ quality of life, fueling a demand for more advanced and costlier bone-related treatments. The global rise in musculoskeletal disorders, such as fractures, scoliosis, osteoporosis, bone infections, tumors, congenital defects, oral pathologies, and rheumatic conditions like osteoarthritis, further complicates the medical landscape [6]. Understanding the biology of bone is the first step toward developing effective treatments for bone lesions. Bone is a nanocomposite comprising inorganic nanocrystalline hydroxyapatite (HA), organic components like collagens, and water, forming a nanostructured extracellular matrix (ECM) crucial for cell adhesion, proliferation, and differentiation [7,8,9].

Despite advancements in bone regeneration techniques, significant challenges remain in orthopaedic practice, with bone being the second most transplanted tissue after blood. Bone implants serve various purposes including fracture repair, joint replacement, filling bone loss gaps, spinal fusion, the correction of congenital bone defects, and as anchors for dental prosthetics [6]. However, bone grafts, including autografts, allografts, and xenografts, are limited by issues such as supply shortages, risk of infection, rejection, and the need for secondary surgeries [6,10]. Metal implants, like titanium and stainless steel, are also problematic due to their non-degradability, rigidity, and poor integration with host tissue [6].

The treatment of bone defects requires a multifaceted approach [10]. In some cases, artificial bone implants made of biocompatible materials are needed [11]. Bone tissue engineering (BTE) strategies (Figure 1) are emerging as a promising approach for replacing lost or damaged bone tissue, typically based on a scaffold support. Scaffolds provide architectural support and homing to cells and bioactive compounds including growth factors (GFs), or provide various stimulation methods (mechanical, electrical, magnetic, acoustic, etc.) that can be applied in bioreactors and in vitro cultures to direct cellular growth in specific orientations [12].

A bio-scaffold is an artificial three-dimensional porous structure constructed from biomaterials such as ceramics, metals, and polymers, mimicking the bone matrix’s 3D repeating unit cells or extracellular matrices (ECMs) [10]. For effective bone regeneration, these scaffolds must possess specific characteristics including the optimal pore architecture and connectivity, appropriate geometry, robust mechanical properties, efficient cell-seeding capabilities, and effective nutrient and metabolite transport [14].

The main requirements for a bio-scaffold, especially for bone regeneration, include the following:Biocompatibility: This ensures that the implant material does not cause adverse reactions inside the body; the materials used must be non-toxic or have low-toxicity properties, and must not cause an immune reaction, and must prevent post-surgical infections; they should support cell attachment, proliferation, and differentiation;Bioactivity, promoting osteointegration: The scaffold should interact positively with the biological environment, promoting new bone tissue. This is achieved through the integration of osteoinductive and osteoconductive properties that promote bone growth;Mechanical properties: The bio-scaffold is engineered to possess the necessary mechanical strength and elasticity that mirrors the mechanical properties of the native bone at the implant site; it should provide structural support until the new tissue is formed and can bear loads;Permeability: Porosity (density and pore size) is relevant for the adequate interconnectivity of pores that allows for cell migration, vascularisation, and nutrient/waste exchange; the pore size is crucial for determining the type of tissue that will infiltrate the scaffold;Degradation rate: The degradation rate of the scaffold material must be carefully aligned with the pace of new tissue growth. This alignment allows for a systematic transition of mechanical loads to the regenerating tissue, thereby supporting seamless tissue integration and structural stability.Manufacturability: The ability to fabricate the scaffold into complex shapes customised to the patient’s defect, using techniques like 3D printing, ensuring a perfect customised fit, and a precision placement, is critical for the implant’s longevity, effectiveness, and integration with the bone;Sterilizability: The scaffold must be sterilizable without losing its structural integrity or biological properties;Durability: This ensures long-term functionality without the need for replacement;Affordability and availability: The materials and technology used should be cost-effective and readily available to ensure that the treatment can be accessed by a wide range of patients;Regulatory compliance: The scaffold must meet the regulatory requirements of medical devices for the specific application it is intended for.

A bio-scaffold is a structure designed and produced to support tissue growth and regeneration in tissue engineering. It acts as a template that guides the formation of new tissue by providing an optimal environment for cellular adhesion, proliferation, and organisation into functional tissue. The success of bone tissue engineering treatments relies on scaffolds’ ability to maintain the viability and functionality of both exogenous and endogenous cells [15]. Bio-scaffolds are typically biocompatible, and their properties can be customised to match the mechanical and biological requirements of the specific tissue being repaired or regenerated. These scaffolds can be fabricated from a diverse array of materials, including biodegradable polymers, bioceramics, metals, and composites. They are also designed to degrade at a predetermined rate, allowing for the gradual replacement by natural tissue over time.

Additive manufacturing (AM) is a new and emergent process. In fact, bone scaffolds’ requirements list can be fulfilled using low-cost FDM technology, due to the increasing capacity to create new filaments of biocompatible and biodegradable materials.

## 2. Bio-Scaffolds for BTE

### 2.1. Bio-Scaffolds Requirements

In orthopaedics, various bio-scaffolds have been explored for their ability to improve cell viability, attachment, proliferation, osteogenic differentiation, integration with host tissue, vascularisation, and load-bearing capacity. A bio-scaffold is a synthetic structure designed to facilitate the formation of three-dimensional tissues [6,7]. These scaffolds can operate either as acellular frameworks or as carriers for cells and/or therapeutic agents. Upon implantation at the injury site, acellular scaffolds are designed to enhance the colonisation of host cells, which is critical for tissue regeneration (Figure 2).

Bio-scaffolds may be employed alongside various cell types that support bone formation in vivo through differentiation into osteogenic lineages or by emitting specific soluble factors. These cells can be cultured and expanded ex vivo prior to their implantation. A bone bio-scaffold can be considered successful if, after implantation, it demonstrates the following capabilities: (a) promoting cell differentiation, proliferation, and migration; (b) facilitating the transfer of biological fluids; (c) possessing adequate surface roughness for tissue adherence; (d) demonstrating strong biocompatibility and biodegradability; and (e) retaining mechanical integrity until complete recovery [1].

Despite advancements, a notable gap persists between laboratory research and clinical implementation, presenting opportunities to tailor scaffold properties to meet distinct biological, clinical, production, economic, and regulatory needs.

For bone tissue engineering (BTE) applications, the ideal bio-scaffold should fulfill criteria across four primary categories: biological, structural, material, and manufacturing technological requirements.

Biological requirements

A bio-scaffold must demonstrate biocompatibility and exhibit non-toxic or low-toxicity properties. It is essential for cells to not only adhere to the scaffold but also to maintain normal functions such as proliferation, differentiation, and new matrix production [16]. Additionally, the scaffold should be bioresorbable and biodegradable to align tissue formation with its degradation. The byproducts of this degradation must be non-toxic and capable of being efficiently eliminated by the body without impacting other organs. In the case of load-bearing implants, it is crucial that the mechanical strength changes during degradation are controlled, allowing for a gradual transfer of the load to the regenerating tissue.

A critical factor to consider is scaffold bioactivity, which refers to its capacity to interact with the adjacent living tissues or organs. Unlike conventional passive biomaterials that may exhibit low or ineffective interaction with their environment, bioactive scaffolds are engineered to promote cell migration, differentiation, tissue neoformation, and integration within the host, all while mitigating adverse effects such as scarring.

Structural requirements

For effective cell penetration, vascularisation, nutrient distribution, and waste removal, bio-scaffolds must possess uniformly distributed, interconnected pores with a high degree of porosity. Another crucial factor that has a significant impact on allowing proper cell colonisation is the average pore size, which varies based on the specific cellular and tissue requirements of the engineering process. Each bio-scaffold type has an optimal pore size range that is crucial for the targeted cell types and tissues [17].

In bone tissue engineering (BTE), the mechanical properties of bio-scaffolds, such as the elastic modulus, tensile strength, fracture toughness, fatigue resistance, and percentage of elongation, are crucial. These properties should be adjusted to match with the properties at the implantation site to minimise complications like stress shielding, implant-induced osteopenia, and the likelihood of re-fracture.

Additionally, surface topography is a vital aspect that can be tailored by integrating artificial extracellular matrices (ECMs) and/or bioactive molecules released post-implantation. This approach not only promotes bone tissue regeneration but also modulates the immune response.

Material requirements

In bone tissue engineering (BTE), a biomaterial functions as a provisional matrix that furnishes a specific environment and structure conducive to bone formation and growth. Commonly utilised biodegradable synthetic polymers include polyhydroxy butyrate (PHB), polylactic acid (PLA), polyglycolic acid (PGA), poly(lactide-co-glycolide) (PLGA), polyethylene glycol (PEG), polyvinylpyrrolidone (PVP), polyhydroxyalkanoate (PHA), polycaprolactone (PCL), poly(l-lactide-co-ε-caprolactone) (PLCL), poly(glycerol sebacate) (PGS), and various synthetic hydrogels [18,19,20,21,22]. Among these, PCL and PLA are the predominant biopolymers chosen for tissue engineering applications. On the other hand, biodegradable natural polymers such as collagens, cellulose, chitosan, and alginate are also used, although synthetic biopolymers tend to be more cost-effective. The chemical characteristics of these polymers are crucial as they govern the degradation process, an integral aspect of scaffold engineering [23].

Figure 3 illustrates a comparison of natural and synthetic biomaterials used in scaffolds.

Manufacturing technologies requirements

In the field of bone regeneration and the production of three-dimensional artificial bone scaffolds, different manufacturing techniques are employed. Traditional methods include solvent casting, particle and salt leaching, freeze-drying, phase separation, fibre bonding, foam gel, molding, and gas foaming.

Three-dimensional printing techniques develop products via layer-by-layer deposition. Scaffolds, on the other hand, are fabricated via a dual-step additive manufacturing (AM) process beginning with an acellular scaffold structure. These are subsequently seeded with cellular structures and cell-laden constructs that mimic natural tissue organisation. Acellular bio-scaffolds are produced using technologies such as fused deposition modelling (FDM), stereolithography (SLA), and selective laser sintering (SLS) [24].

Additive manufacturing (AM) has become an alternative to traditional methods, enabling an integration with computer systems for the systematic design and the production of reproducible structures suitable for tissue engineering. The AM technologies generally used in tissue engineering (TE) include material extrusion, where material is deposited through a nozzle or orifice in successive layers to form scaffolds. The vat photopolymerisation method uses light to cure photopolymer resin in a vat. Layers are selectively cured by a light source, usually ultraviolet light (UV), creating precise and complex scaffold structures. In powder bed fusion, a laser or electron beam fuses powder particles together to form a solid structure. It is highly versatile, allowing the use of various materials, including metals and polymers. The material jetting technique deposits droplets of material from a print head, which then solidify to form layers of the scaffold. It allows for a high precision and can use multiple materials simultaneously. Direct writing systems use a computer-controlled mechanism to deposit material directly onto a substrate. This method allows for a high precision and customisation in scaffold fabrication. Figure 4 illustrates the conventional fabrication and AM technologies employed in TE applications [25,26].

### 2.2. Bio-Scaffolds Design and Development

#### 2.2.1. BTE Scaffold-Based Strategy Development

Developing a BTE scaffold-based strategy involves addressing several key issues (Figure 5). The initial step is to identify the appropriate scaffold-based BTE therapy, which includes selecting material properties and manufacturing methods, and determining whether to use a multi-component or single-component treatment.

The development of a BTE scaffold-based strategy necessitates addressing a series of critical steps. These steps range from pre-clinical studies, both in vitro and in vivo, to obtaining necessary approvals for clinical trials, followed by the commercialisation of the scaffolds and managing the expectations of patients. Each phase of this elaborate process requires authorisation from national and international regulatory bodies. Key participants in this intricate ecosystem include surgeons who place these scaffold-based constructs during clinical trials, biomaterials scientists who develop the scaffolds, companies that manufacture these structures for widespread commercial distribution, researchers who perform pre-clinical testing, and the regulatory authorities, along with the patients who have specific expectations from these treatments [8].

#### 2.2.2. Scaffold Design Cause–Effect Diagram

Figure 6 illustrates the essential requirements and properties of scaffolds. The primary design criteria for optimal scaffolds include biocompatibility, biodegradability, mechanical strength, and interconnected porosity. Scaffolds are also required to support a hierarchical structure and provide a biomechanical and physiological microenvironment that minimises immune responses and facilitates the replacement of the scaffold with regenerated tissues. Furthermore, these structures should promote cellular infiltration and migration, as well as the efficient diffusion of nutrients [27].

## 3. Manufacturing of Scaffolds by FDM

### 3.1. Additive Manufacturing

In comparison to conventional fabrication techniques like porogen leaching, phase-separation/freeze-drying, and gas-foaming, scaffolds produced through 3D printing offer substantial improvements. Traditional methods typically generate scaffolds with heterogeneous properties, wide variations in pore size, suboptimal pore interconnectivity, and compromised mechanical characteristics. Additive manufacturing (AM), on the other hand, allow the precise and consistent production of scaffolds with engineered architectures based on computer designs.

The field of regenerative scaffolds for orthopaedic use has been transformed by AM, which permits the customisation of scaffold geometries and material formulations to closely replicate bone structure. However, the literature presents mixed findings regarding the most effective geometric designs and material formulations.

AM technologies are adept at swiftly crafting scaffolds with intricate external contours and internal porous frameworks. To construct an effective 3D porous structure, it is essential that the design supports both three-dimensional bone healing and vascularisation within the healing site [28].

Due to the AM quick turnaround time in the field of customised implants, it shows great prospects in orthopaedic application.

AM fabricates the final three-dimensional object from a digital CAD model by depositing the material layer-by-layer, utilising either liquid, solid, or powder bases. Numerous AM techniques, including fused deposition modelling (FDM), selective laser sintering (SLS), selective laser melting (SLM), stereolithography (SLA), electron beam melting (EBM), three-dimensional printing, bioprinting, and electrospinning, have proven effective in creating complex bone implants [8,29].

AM provides meticulous control over scaffold porosity, pore size, and the mechanical and chemical properties, enhancing the ability to emulate natural bone structure. Moreover, these technologies allow for the customisation of material composition within the scaffold’s surface, interface, or volume.

While FDM is used to create sustainable engineering products, customised implants, and advanced biomedical devices, the mechanical strength and durability of FDM-produced items remain inferior to those produced by conventional methods, limiting their broader application. To address this, the integration of short fibres—either natural or synthetic—and advanced nanomaterials to fortify the polymer matrix has notably increased, improving the load-bearing capabilities of FDM-printed components.

### 3.2. FDM Process

A comprehensive description of FDM processing parameters is crucial, as these parameters significantly influence the quality of the printed product. An FDM 3D printer operates by the deposition of melted filament material over a platform layer-by-layer (Figure 7). In the fused deposition modelling (FDM) process, molten thermoplastic material is extruded through a nozzle and deposited on a build platform in a pathway that is pre-defined by computer-aided design (CAD) and computer-aided manufacturing (CAM) systems [30,31]. The extrusion nozzle is mounted on a three-axis system, allowing movement along the X, Y, and Z axis. Each layer is completed on the XY plane before the platform descends along the Z axis to start the next layer. Upon deposition, the material undergoes cooling and solidification.

The parameters of material deposition, such as the amount of material extruded, the distance between the deposition paths, and the layer height on the Z axis, can be finely tuned. This allows for the production of 3D scaffolds with a controlled pore size and porosity. The key benefits of FDM include its ability to achieve high porosity in materials, robust mechanical strength, the elimination of the need for toxic solvents, and versatility in material handling and processing [32].

FDM systems present several advantages, including: (1) the ability to fabricate without geometric constraints, (2) reduced costs of technology and materials, (3) straightforward operation and material management, (4) functionality at low temperatures, (5) decreased production and maintenance costs, (6) minimal toxicity of processes, (7) low energy requirements, (8) the use of various material systems, (9) the capability to produce coloured components, (10) compact design and suitability for office settings, (11) quiet operation with minimal dust production, (12) minimal odour emission, (13) the potential for mass customisation, and (14) the ability to personalise products. However, FDM systems also have disadvantages, such as: (1) the low production speed; (2) the compromised accuracy and resolution; (3) the limited surface finish; (4) issues like the staircase effect, distortion, shrinkage, and warping; (5) the need for support structures in complex geometries; (6) the subsequent requirement to remove these supports; (7) a restricted selection of materials; (8) the limited mechanical strength of the parts; and (9) the limited build volume or workspace [9].

The precision with which the FDM process maintains dimensional stability in fabricated components is crucial for its acceptance in the market. The dimensional accuracy of products created via FDM is affected by numerous conflicting parameters, as illustrated in Figure 8 [33,34,35]. This complexity necessitates the careful management of the operating conditions and parameters to optimise the output quality.

A primary challenge associated with the FDM technique lies in the requirement for preformed filaments that consistently exhibit uniform dimensions and material properties, which are essential for their effective passage through the rollers and nozzle [15].

## 4. Materials for Printed Bio-Scaffolds

Most bio-scaffolds employed in bone tissue engineering (BTE) applications are currently derived from polymers, bioactive ceramics, and their composites. Bioactive ceramics, sourced either naturally or synthetically, include materials such as coralline, hydroxyapatite (HA), tricalcium phosphate (TCP), sulphate, bioactive glass (BG), and calcium silicate. These materials are chemically akin to bone and demonstrate significant compressive strength but limited ductility, which contributes to their high resistance to deformation and inherent brittleness. Additionally, these ceramics present challenges in processing. The roles of polymers and composite materials in bio-scaffolds will be explored further below.

### 4.1. Polymers

Polymers are categorised into natural and synthetic types, as illustrated in Figure 3. Polymers derived from natural sources such as fibrin, hyaluronic acid, chitosan, and collagen are known for their excellent biocompatibility, osteoconductivity, and minimal immunogenicity. However, they exhibit several drawbacks, including uncontrollable degradation rates, low mechanical stability, considerable variability in their properties, and several processing challenges. Conversely, synthetic polymers like polylactic acid (PLA), polycaprolactone (PCL), polyanhydride, polypropylene fumarate (PPF), and polyphosphazene, and poly(glycolic acid) (PGA) are advantageous due to their predictable degradation rates (adjustable to synchronise with tissue regeneration), the ability to tailor mechanical properties to bone, and the flexibility to create complex structures conducive to cell attachment.

Figure 9 illustrates the predominant use of polycaprolactone (PCL) and polylactic acid (PLA) as biopolymers for tissue engineering (TE) applications. These synthetic polymers are distinguished by their economic viability relative to their natural biodegradable polymers and the capability to engineer their macrostructures with precision. The intrinsic chemical structures of PCL and PLA are crucial in determining their degradation pathways, which is vital for the development and manufacturing of scaffolds. The degradation kinetics of these polymers are critically important, as they directly impact the timing of tissue integration and regeneration. Consequently, tailoring these structures allows for the fine-tuning of scaffold properties to match the dynamic mechanical and biological requirements of the target tissue [27].

Synthetic polymers can be produced economically, in consistent large volumes, and with an extended shelf life. However, they face a notable limitation: their interaction with cells is less effective compared to natural polymers, which possess inherently superior bioactive characteristics. A crucial category of polymers in bone tissue engineering (BTE) is hydrogels. These hydrophilic polymer networks are capable of absorbing water from as little as 10 to 20% to several thousand times their dry weight. This absorption capability facilitates cellular adhesion, proliferation, and differentiation. Hydrogels, whether natural (such as agarose, alginate, and gelatin) or synthetic (including those based on polyvinyl alcohol and polyacrylates), are adept at emulating the topography of the extracellular matrix (ECM).

### 4.2. Hybrids and Main Functional Fillers

Hybrid materials combine two or more substances, each bringing unique benefits and limitations to the resulting composite. These hybrids may consist of copolymers, polymer blends, or polymer-based composites with metals or ceramics. Copolymers, such as PLGA—which is a copolymer of polylactide and polyglycolide—stand out in bone tissue engineering (BTE) for their biodegradability and manufacturing ease. Blends of polymers, like the PLGA-polyphosphazene mixture, address the challenge of acidic byproducts from PLGA degradation which can cause tissue damage and implant failure; polyphosphazenes degrade into neutral or basic products; thus, the blend yields nearly neutral degradation byproducts. Polymer–ceramic composites are particularly biomimetic, reflecting the natural composition of bone, which is an organic–inorganic composite of hydroxyapatite crystals and collagen fibres. Such composites have proven highly effective in bone regeneration, surpassing the performance of their individual components. The inclusion of inorganic elements like bioceramic and bioglass particles, carbon nanotubes, or magnesium metal or alloy particles can enhance the mechanical properties and other characteristics of scaffolds.

Figure 10 illustrates a schematic representation of the composite filament extrusion process utilised in fused deposition modelling. This process begins with the feeding of composite pellets—comprising polymers and fillers—into an extruder. The extruder processes these pellets into composite feedstock, which is then spooled. The spooled feedstock is used in a 3D printer, FDM, to create printed samples. These samples are subsequently subjected to mechanical testing and analysis to evaluate their properties.

Composites can be classified as biodegradable or non-biodegradable [16]. Figure 11 provides an overview of the main types of biodegradable and non-biodegradable materials used in FDM.

### 4.3. Material Cause–Effect Diagram

In the production of porous artificial bone scaffolds, critical for the recovery of injured tissues, it is essential to use bioactive and biocompatible materials that allow for controlled degradation and resorption rates.

The materials chosen for FDM are typically polymer-based, exhibiting different physical, mechanical, and thermal characteristics. The selection of these polymers is tailored to meet the specific needs and requirements of the application. The quality and properties of these polymers can be improved by incorporating fillers like ceramics, nanoparticles, and wood fibre.

In many orthopaedic regenerative engineering applications, PLA-based biomaterials are considered as the benchmark, attributed to their versality in manufacturing, biodegradability, and compatibility with biomolecules and cells.

Figure 12 represents a cause–effect diagram that identifies significant factors influencing the extensive use of biodegradable polymers in tissue regeneration [27].

Scaffolds based on biomaterials for bone regeneration need to have the following characteristics: (1) they must be biodegradable, biocompatible, biofunctional, and non-toxic; (2) they should provide sites for cellular adhesion to facilitate cell attachment and proliferation, making them osteoconductive; (3) their degradation rates should align with the rates of tissue regeneration; (4) they require chemical and physical properties that mimic native bone tissue; (5) they require a highly interconnected three-dimensional pore network with pores of sufficient size to support cell proliferation and effective nutrient and waste exchange; (6) they need the capability to incorporate essential elements like growth factors, stem cells, and anti-inflammatory agents for osteogenicity; (7) they must promote robust adhesion and integration with newly formed bone tissue, thus supporting osteointegration; and (8) they should have the capability to attract progenitor cells, support their differentiation into specific lineages, and promote new bone formation, thus being osteoinductive [36].

## 5. Scaffold Design

### 5.1. Porosity

Permeability is a critical factor in scaffold design as it impacts both the mechanical strength and the biological functionality, including cell infiltration, nutrient distribution, and vascularisation. The porosity of scaffolds created through FDM is a vital factor in tissue engineering by affecting both the physical and biological performance of the scaffold. An optimally designed scaffold should support tissue regeneration while preserving the necessary physical and mechanical characteristics. However, replicating the complex architecture of natural bone poses significant challenges in creating artificial bone scaffolds [25].

Scaffold design, which incorporates the appropriate geometry and architecture, is crucial for providing sufficient mechanical support and modulating cellular activities. Insights from developmental biology inform the underlying mechanisms of tissue formation, offering valuable guidance for tissue repair and regeneration.

Porous scaffolds aim to replicate the extracellular matrix (ECM) of tissues. The porosity of these scaffolds, including aspects such as pore size, shape, distribution, and connectivity, affects both the mechanical strength and the efficiency of mass transfer. These scaffolds are designed to facilitate cell distribution, adhesion, migration, and nutrient and waste exchange. Scaffold permeability, influenced by its pore dimensions, orientation, and interconnectivity, is a critical factor that can predict mass transport dynamics and material degradation kinetics. Porous scaffolds serve as three-dimensional artificial environments that balance mechanical integrity, mass function, and biofactor integration during tissue growth. Figure 13 illustrates the development process of porous scaffolds for tissue engineering, beginning with design considerations at the cellular and tissue levels, proceeding through fabrication methods, moving to characterisation techniques post-manufacture, and ending with a review of the current perspectives and clinical applications of these scaffolds [37].

Scaffold design is initiated through the systematic repetition of unit cells specialized in software. The process begins by conceptualising a unit cell that encapsulates desired characteristics such as porosity, mechanical strength, and biocompatibility. Through a custom scaffolds generator software, these unit cells are modelled in three dimensions [30,38]. The software’s advanced capabilities permit the precise manipulation of pore size, shape, and arrangement, which are critical for enhancing cellular adhesion, proliferation, and differentiation. After the unit cell design is finalized, it is replicated in a periodic pattern to form the larger scaffold structure. This approach guarantees the uniformity and consistency essential for the successful integration with host tissue.

The final fabrication step in scaffold involves using 3D printing technologies like fused deposition modelling (FDM). These technologies are highly compatible with the designs generated in software, allowing for high-precision fabrication that preserves the integrity of the design parameters. FDM enables the layer-by-layer construction of heterogeneous scaffolds structures with polyhedrons directly from digital models, ensuring a high degree of fidelity between the designed and manufactured product. The capability to print different materials in a single scaffold allows for the creation of regions with distinct properties. One of the significant advantages of using polyhedral structures in scaffolds is their high surface-to-volume ratio, which increases the available surface area for cell attachment and growth. Moreover, the interlocking nature of polyhedrons can enhance the mechanical stability of the scaffold [25].

### 5.2. Scaffold Design Cause–Effect Diagram

The design and production of scaffolds are dictated by evaluations of diverse properties, such as mechanical, biological, and physicochemical characteristics, tailored to meet specific feasibility criteria and requirements. Key factors that play a decisive role in the fabrication of scaffolds include the interconnectivity of pores, their dimensions and shape, the overall porosity, structural integrity, and the rate of degradation [39]. Porosity, density, and size critically define the structural and functional attributes of materials. The optimal pore size is essential for facilitating cell infiltration and nutrient exchange, ensuring adequate biological integration. In the design of bone scaffolds, a number of critical parameters are key to attaining optimal functionality. The spatial arrangement of pores throughout the scaffolds significantly impacts its mechanical behaviour. There are three common configurations of pore distribution: (1) crosswise, with pores oriented perpendicular to the scaffold axis; (2) lengthwise, with pores aligned parallel to the scaffold axis; and (3) eccentric, with pores distributed irregularly. The shape of individual pores affects scaffolds’ properties. Common pore geometries include: (1) triangular, (2) circular, (3) square, (4) rectangular, (5) wiggle, and (6) honeycomb. Each shape affects parameters such as mechanical load distribution and cellular adhesion, playing a critical role in the scaffold’s efficacy. The surface area of pores impacts cell infiltration, nutrient diffusion, and the rate of scaffold degradation. Distinct categories based on pore size have been established: (1) small (surface area of 0.25 mm^2^), (2) medium (surface area of 0.5625 mm^2^), and (3) large (surface area of 1 mm^2^). Properly selecting the pore size is crucial for optimising the process for tissue regeneration.

A summary diagram illustrating the different steps required in scaffold manufacturing is presented in Figure 14 [40].

### 5.3. Scaffold Design Approach

A V-diagram strategy is suggested for the development of the bio-scaffold in Figure 15. Five areas can be identified for the development of an ideal scaffold:Project definition: This initial phase involves defining the specific requirements and objectives of the scaffold, including its intended medical applications, target tissue type, and the essential properties it must possess;Material phase: In this stage, the selection and formulation of the materials take place. The materials chosen are typically biocompatible and biodegradable to ensure they do not evoke an adverse reaction in the body and can degrade at a rate compatible with tissue regeneration. The development will focus on biopolymers that integrate well with biological systems without necessitating subsequent surgical removal;Structural design: This phase involves detailed modelling on the scaffold’s architecture. Critical parameters such as pore size, shape, interconnectivity, and the overall three-dimensional structure are designed to meet the specific mechanical and biological needs of the target tissue. Advanced CAD software may be used to achieve precise design specifications and optimize the scaffold’s structural integrity;The scaffold manufacturing phase: Utilising technologies like 3D printing—specifically, fused deposition modelling—the scaffold is fabricated. This phase focuses on accurately replicating the design specifications in the physical model, ensuring that the scaffold maintains its functional integrity under physiological conditions;The testing and integration of the scaffold through a case study: The final phase involves the rigorous testing of the scaffold in both in vitro and in vivo settings to evaluate its biocompatibility, mechanical stability, and efficacy in promoting tissue regeneration. Integration trials through case studies or clinical trials assess the scaffold’s performance in a real-world medical scenario, confirming its readiness for surgical use and integration within the human body.

Scaffolds are made with an optimized composite (mixture of biocompatible and biodegradable materials and an ideal filler previous studied). The biocompatible, biodegradable, and, probably, biopolymer material will be a focus of development because, when this bio-scaffold is used in medical surgery, no additional surgery will be required to remove the implant.

## 6. Concluding Remark

Despite rapid advancements in BTE, several challenges remain. It is a very important goal to define clear objectives for future research and development, including:The integration of materials and technologies: There is a pressing need to integrate diverse materials and technologies to develop scaffolds that effectively facilitate human bone healing. This includes the combination of different biopolymers, ceramics, and bioactive substances that should aim to replicate the intricate structure and functionality of natural bone;The prioritisation of low-cost, sustainable, and biocompatible materials: Developing scaffolds using materials that are not only cost-effective and sustainable but also biocompatible is vital. These materials should promote tissue compatibility without eliciting any adverse immune responses, thus making treatments accessible and safe for a broader population;Selecting the adequate composite for the optimal scaffold: The choice of the right composite is critical. These composites should possess the necessary mechanical strength, support cell attachment and growth, and degrade at a rate congruent with new tissue formation, thereby providing structural integrity during the healing process;Designing the scaffold in order to optimize the structural properties, degradation, and healing: The structural design of the scaffold must be optimized to enhance its mechanical properties, control degradation rates, and support the healing process. This involves the precise engineering of the scaffold’s architecture, such as pore size, shape, and interconnectivity, which are crucial for vascularisation and nutrient diffusion;The in situ cell growth: Encouraging cell growth directly within the scaffold at the site of implantation represents a significant frontier in BTE. Developing scaffolds that can not only support but also stimulate in situ cell proliferation and differentiation is essential for successful integration and healing.

## Figures and Tables

**Figure 1 bioengineering-11-00769-f001:**
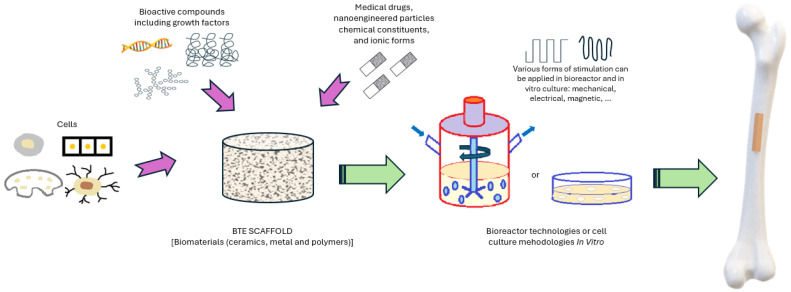
Bone tissue engineering strategies (adapted from [13]).

**Figure 2 bioengineering-11-00769-f002:**
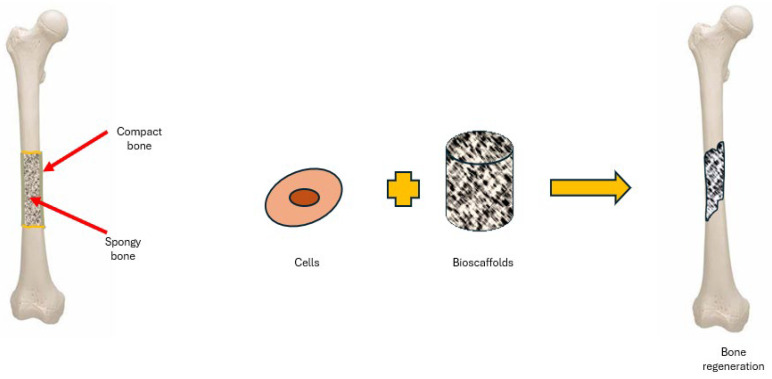
The hierarchical pore structure of bone tissue.

**Figure 3 bioengineering-11-00769-f003:**
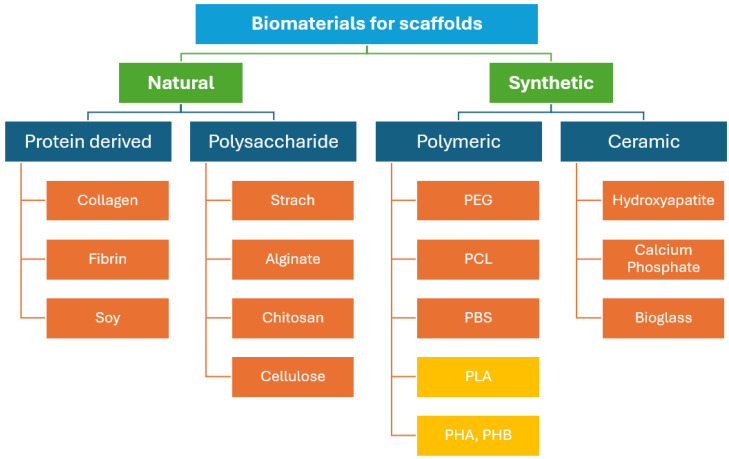
Natural and synthetic biomaterials for scaffolds.

**Figure 4 bioengineering-11-00769-f004:**
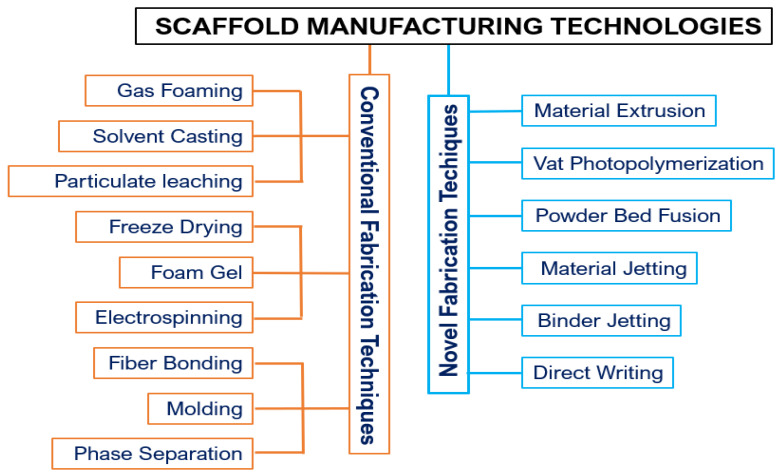
Scaffold manufacturing technologies.

**Figure 5 bioengineering-11-00769-f005:**
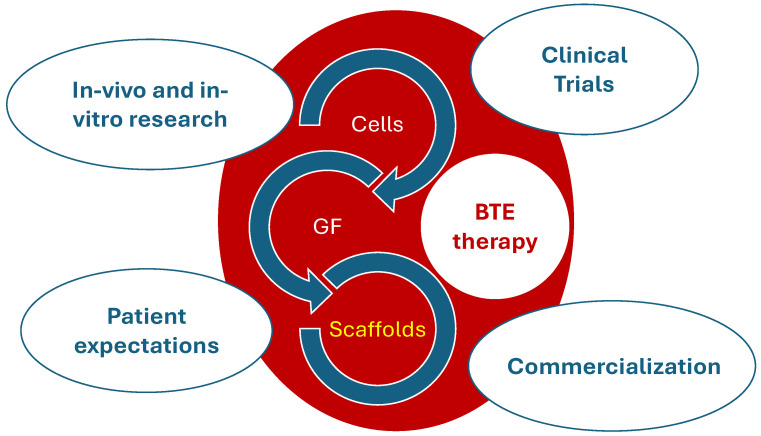
Bone tissue engineering (BTE) scaffold-based strategy development. The central elements include cells, growth factors (GFs), and scaffolds, which interact dynamically in the BTE process. The broader context includes in vivo and in vitro research, clinical trials, commercialisation, and patient expectations, highlighting the multifaceted approach required for successful BTE therapy development and implementation.

**Figure 6 bioengineering-11-00769-f006:**
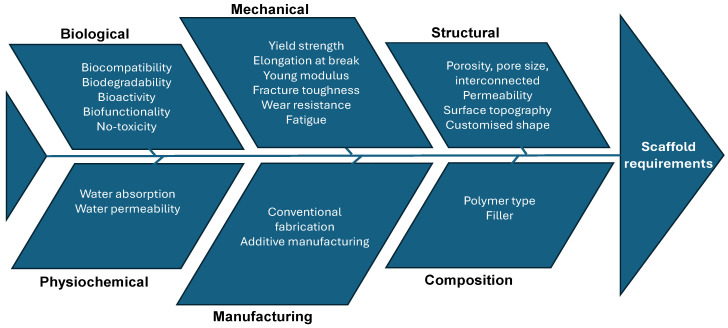
Cause-and-effect diagram of scaffolds requirements.

**Figure 7 bioengineering-11-00769-f007:**
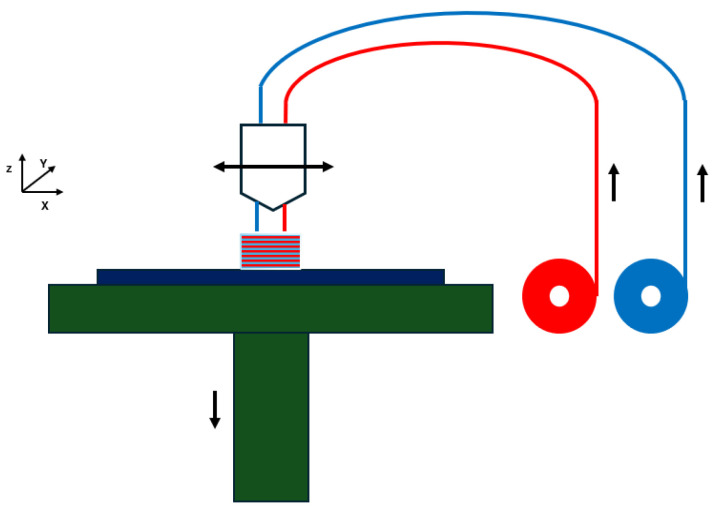
The principle of FDM method.

**Figure 8 bioengineering-11-00769-f008:**
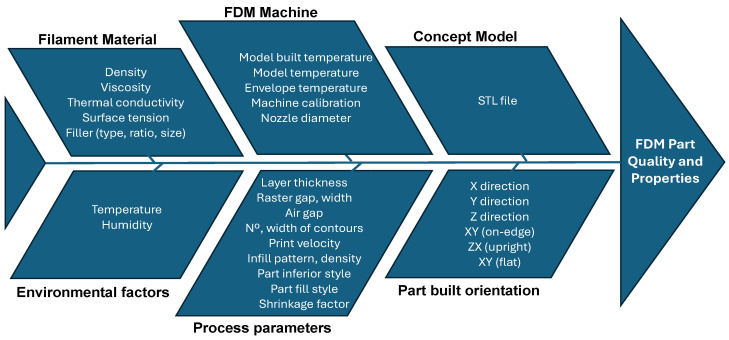
Cause-and-effect diagram of FDM process parameters.

**Figure 9 bioengineering-11-00769-f009:**
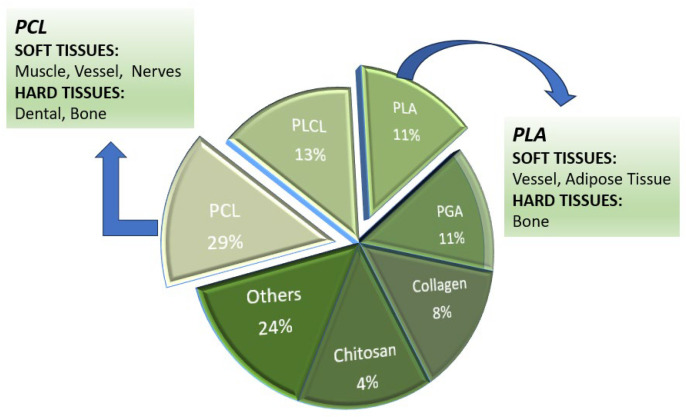
A selection of biopolymers used in scaffold production for bone tissue engineering applications.

**Figure 10 bioengineering-11-00769-f010:**
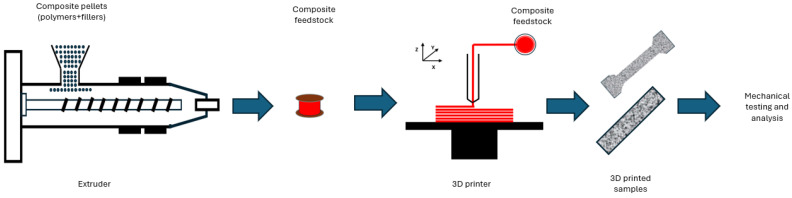
Diagram illustrating the composite filament extrusion process.

**Figure 11 bioengineering-11-00769-f011:**
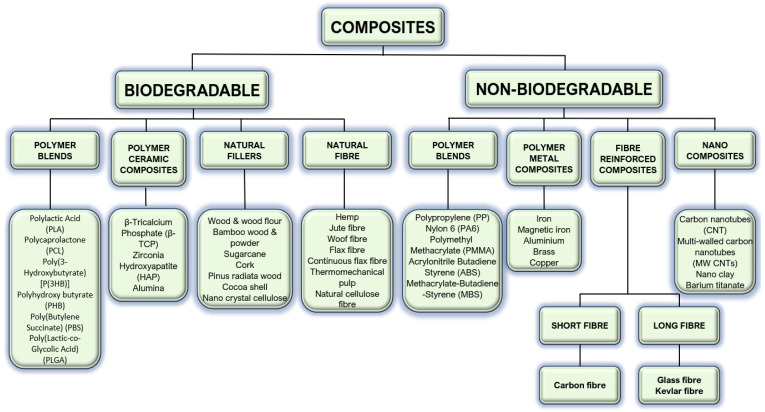
Biodegradable and non-biodegradable materials used in FDM.

**Figure 12 bioengineering-11-00769-f012:**
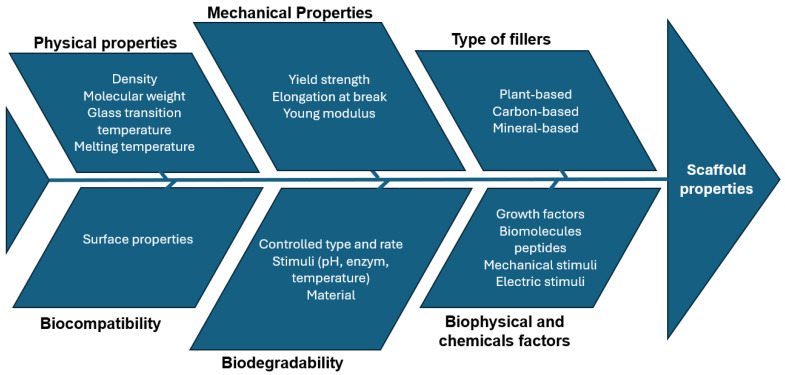
Material cause–effect diagram.

**Figure 13 bioengineering-11-00769-f013:**
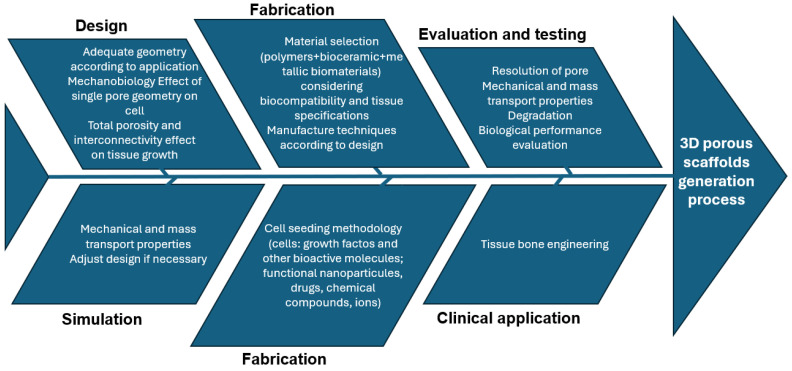
The process of developing porous scaffolds.

**Figure 14 bioengineering-11-00769-f014:**
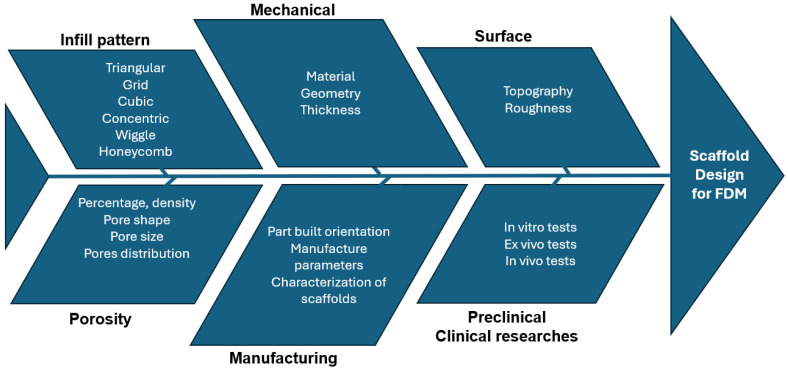
Scaffold design cause–effect diagram.

**Figure 15 bioengineering-11-00769-f015:**
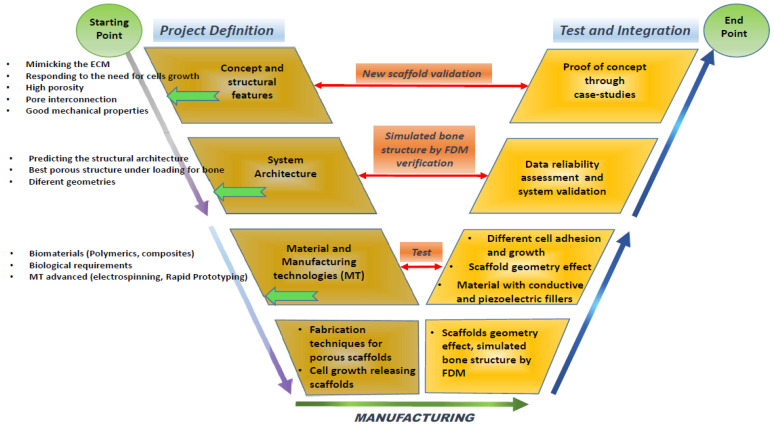
V-diagram approach for the development of an optimal scaffold.

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
