# Peer review of "On the Fused Deposition Modelling of Personalised Bio-Scaffolds: Materials, Design, and Manufacturing Aspects"

_bioengineering, 2024, doi:10.3390/bioengineering11080769_

Round 1
Reviewer 1 Report
Comments and Suggestions for Authors
Dear Authors,
Bone tissue engineering aims to heal bone defects and replace impaired tissues using 3D printing, especially fused deposition modeling (FDM), to create biocompatible and biodegradable scaffolds. These scaffolds may be to improve support cell adhesion, proliferation, and differentiation, with essential properties like porosity, permeability, and degradation influencing cellular behavior. Polylactic Acid (PLA) and Polycaprolactone (PCL) are commonly used. This research optimizes these materials and FDM technology to produce effective bone healing scaffolds.
Reported below are my suggestions:
-Add a graphical Abstract in order to summarize the topic of the review
-In Figure 1, please clarify the meaning of the blue arrows. Additionally, in the same figure, it is not completely clear if the box “Various forms of stimulation…….”. referred to both bioreactor and /or in vitro culture.
- In Figure 3, check the word “collagen”
-I suggest adding a caption that describes each figure. And in case there are some acronyms, for example, in Figure 5 “GF,” please report the meaning in the caption
-In Figure 4, the authors mention the novel techniques for scaffold manufacturing (material extrusion, vat photopolymerization, powder bed fusion, material jetting, and direct writing systems) without furnishing a short description for each one.
-In Figure 9, I suggest moving the title “PLA/PCL in tissue engineering” to the upper position or removing
I suggest adding a short description of the process represented in the caption of Figure 10 or in the text at line 381.
Finally, to conclude this revision, it should be interesting to know your vision based on the literature about the development of scaffolds in bone tissue regeneration.
Best Regards
Comments on the Quality of English LanguageThe article is fluent and well-written.
Author Response
We would like to express our appreciation to the reviewers for their very useful comments to our work. To resubmit the paper we made the appropriate corrections to answer the reviewers concerns. All the issues are discussed below:
Reviewer #1
Dear Authors,
Bone tissue engineering aims to heal bone defects and replace impaired tissues using 3D printing, especially fused deposition modeling (FDM), to create biocompatible and biodegradable scaffolds. These scaffolds may be to improve support cell adhesion, proliferation, and differentiation, with essential properties like porosity, permeability, and degradation influencing cellular behavior. Polylactic Acid (PLA) and Polycaprolactone (PCL) are commonly used. This research optimizes these materials and FDM technology to produce effective bone healing scaffolds.
Reported below are my suggestions:
-Add a graphical Abstract in order to summarize the topic of the review
Thanks for the suggestion. A graphical abstract was included.
-In Figure 1, please clarify the meaning of the blue arrows. Additionally, in the same figure, it is not completely clear if the box “Various forms of stimulation…….”. referred to both bioreactor and /or in vitro culture.
We made some changes in Figure 1 and add some text in lines 94 and 95: “(…) factors (GF), or provide various forms of stimulation (mechanical, electrical, magnetic, acoustic, …) who can be applied in bioreactor and in vitro culture and help them to grow in a particular direction [12]”
- In Figure 3, check the word “collagen”
Figure 3 was corrected
-I suggest adding a caption that describes each figure. And in case there are some acronyms, for example, in Figure 5 “GF,” please report the meaning in the caption
Figure 5 legend is new. GF was introduced in line 93, 94 “(…) growth factors (GF) (…)”.
-In Figure 4, the authors mention the novel techniques for scaffold manufacturing (material extrusion, vat photopolymerization, powder bed fusion, material jetting, and direct writing systems) without furnishing a short description for each one.
Some description was added in section 2.1 (between lines 229-240): “The AM technologies generally used in Tissue Engineering (TE) include material extrusion - this technique involves the deposition of material through a nozzle or orifice, layer by layer, to create a scaffold. Vat photopolymerization method uses light to cure photo-polymer resin in a vat. Layers are selectively cured by a light source, usually UV, creating precise and complex scaffold structures. In powder bed fusion a laser or electron beam fuses powder particles together to form a solid structure. It is highly versatile, allowing the use of various materials, including metals and polymers. Material jetting technique de-posits droplets of material from a print head, which then solidify to form layers of the scaffold. It allows high precision and can use multiple materials simultaneously. Direct writing systems use a computer-controlled mechanism to deposit material directly onto a substrate. This method allows for high precision and customization in scaffold fabrication.”
-In Figure 9, I suggest moving the title “PLA/PCL in tissue engineering” to the upper position or removing
Our option was remove the title.
I suggest adding a short description of the process represented in the caption of Figure 10 or in the text at line 381.
A short description was added in section 4.2 (in the new version lines 403-407): “It begins with composite pellets, consisting of polymers and fillers, being fed into an ex-truder. The extruder processes these pellets into composite feedstock, which is then spooled. The spooled feedstock is used in a 3D printer, FDM, to create printed samples. These samples are subsequently subjected to mechanical testing and analysis to evaluate their properties.”
Finally, to conclude this revision, it should be interesting to know your vision based on the literature about the development of scaffolds in bone tissue regeneration.
In our opinion, the “concluding remark” section summarizes our vision. In fact, is possible to manufacture FDM scaffolds with biocompatible and biodegradable materials and with porosity and permeability proper to different bone tissues.
Thanks for all suggestions.
Best Regards

Reviewer 2 Report
Comments and Suggestions for Authors
Comments of bioengineering-3029344-
The main weaknesses of the manuscript:
1. What is the gap being addressed by this paper? This should be clearly stated in the last paragraph of the introduction section.
2. The novelty of the work should be clearly described. What benefits could be achieved from this process? Please revise accordingly.
3. The structure of the paper is not adequate. If you decide to use a results section and a discussion section, please do not put any discussion in the results and do not put any new results in the discussion. Your paper should be in the best possible way for the reader to understand.
4. The conclusion should be more concise in a better way.
Author Response
We would like to express our appreciation to the reviewers for their very useful comments to our work. To resubmit the paper we made the appropriate corrections to answer the reviewers concerns. All the issues are discussed below:
Reviewer #2
The main weaknesses of the manuscript:
- What is the gap being addressed by this paper? This should be clearly stated in the last paragraph of the introduction section.
This is a Review paper. The main gap addressed in this paper is the state of art in FDM in order to produce bio-scaffolds. This sentence was added to the end of abstract: “FDM can be used to produce low-cost bone scaffolds, with suitable porosity and permeability.”
- The novelty of the work should be clearly described. What benefits could be achieved from this process? Please revise accordingly.
This is a Review paper. However, we believe, based on our knowledge and of the review analysis that is possible to produce bone scaffolds with FDM technology. This sentence was added to the end of introduction: “Additive manufacturing is a new and emergent process. In fact, bone scaffolds requirements list can be fulfilled using low cost FDM technology, due to the increasing capacity to do new filaments of biocompatible and biodegradable materials.”
- The structure of the paper is not adequate. If you decide to use a results section and a discussion section, please do not put any discussion in the results and do not put any new results in the discussion. Your paper should be in the best possible way for the reader to understand.
This is a Review paper. We do not have a “results” section and neither a “discussion” section. Sections are related with scaffolds, FDM, materials, design. In the end with have a “concluding remark” section.
- The conclusion should be more concise in a better way.
In our opinion, the “concluding remark” section summarizes our vision. In fact, is possible to manufacture FDM scaffolds with biocompatible and biodegradable materials and with porosity and permeability proper to different bone tissues.
Thanks for all suggestions.

Round 2
Reviewer 1 Report
Comments and Suggestions for Authors
Dear Authors,
thank you to have revised all the points.
Best Rergards
Comments on the Quality of English LanguageMinor editing of the English language